

# A Sulfur Dioxide Covariance-Based Retrieval Algorithm (COBRA): application to TROPOMI reveals new emission sources

**Nicolas Theys[1], Vitali Fioletov[2], Can Li[3,4], Isabelle De Smedt[1], Christophe Lerot[1], Chris McLinden[2], Nickolay Krotkov[3], Debora Griffin[2], Lieven Clarisse[5], Pascal Hedelt[6], Diego Loyola[6], Thomas Wagner[7], Vinod Kumar[7], Antje Innes[8], Roberto Ribas[8], François Hendrick[1], Jonas Vlietinck[1], Hugues Brenot[1], Michel Van Roozendael[1]**

[1] Royal Belgian Institute for Space Aeronomy (BIRA-IASB), Brussels, Belgium.

[2] Air Quality Research Division, Environment and Climate Change Canada, Toronto, Canada.

[3] Atmospheric Chemistry and Dynamics Laboratory, NASA Goddard Space Flight Center, Greenbelt, MD, USA.

[4] Earth System Science Interdisciplinary Center, University of Maryland, College Park, MD, USA.

[5] Université libre de Bruxelles (ULB), Spectroscopy, Quantum Chemistry and Atmospheric Remote Sensing (SQUARES), C. P. 160/09, Brussels, Belgium.

[6] Institut für Methodik der Fernerkundung (IMF), Deutsches Zentrum für Luft und Raumfahrt (DLR), Oberpfaffenhofen, Germany.

[7] Max Planck Institute for Chemistry (MPIC), Hahn-Meitner-Weg 1, 55128 Mainz, Germany.

[8] European Centre for Medium-Range Weather Forecast (ECMWF), Shinfield Park, Reading, RG2 9AX, UK.

*Correspondence to:* N. Theys (theys@aeronomie.be)





## ABSTRACT

Sensitive and accurate detection of sulfur dioxide (SO$_2$) from space is important for monitoring
and estimating global sulfur emissions. Inspired by detection methods applied in the thermal
infrared, we present here a new scheme to retrieve SO$_2$ columns from satellite observations of
ultraviolet back-scattered radiances. The retrieval is based on a measurement error covariance
matrix to fully represent the SO$_2$-free radiance variability, so that the SO$_2$ slant column density is
the only retrieved parameter of the algorithm. We demonstrate this approach, named COBRA, on
measurements from the TROPOspheric Monitoring Instrument (TROPOMI) aboard the Sentinel-
5 Precursor (S-5P) satellite. We show that the method reduces significantly both the noise and
biases present in the current TROPOMI operational DOAS SO$_2$ retrievals. The performance of this
technique is also benchmarked against that of the Principal Component Algorithm (PCA)
approach. We find that the quality of the data is similar and even slightly better with the proposed
COBRA approach. The ability of the algorithm to retrieve SO$_2$ accurately is also further supported
by comparison with ground-based observations. We illustrate the great sensitivity of the method
with a high-resolution global SO$_2$ map, considering two and a half years of TROPOMI data. In
addition to the known sources, we detect many new SO$_2$ emission hotspots worldwide. For the
largest sources, we use the COBRA data to estimate SO$_2$ emission rates. Results are comparable
to other recently published TROPOMI-based SO$_2$ emissions estimates, but the associated
uncertainties are significantly lower than with the operational data. Next, for a limited number of
weak sources, we demonstrate the potential of our data for quantifying SO$_2$ emissions with a
detection limit of about 8 kt yr$^{-1}$, a factor of 4 better than the emissions derived from the Ozone
Monitoring Instrument (OMI). We anticipate that the systematic use of our TROPOMI COBRA
SO$_2$ column data set at a global scale will allow identifying and quantifying missing sources, and
help improving SO$_2$ emission inventories.

## 1. INTRODUCTION

Sulfur dioxide (SO$_2$) in the atmosphere rapidly oxidizes into sulfuric acid and sulfate aerosols,
which have environmental effects ranging from local and long-range air pollution to global climate
impact. SO$_2$ is released into the atmosphere from anthropogenic activities, due to fossil fuel



burning (coal, oil and gas) and smelting, and from natural sources, mainly volcanoes. Satellites
provide a viable means to monitor global $SO_2$ emissions and assess their environmental impacts.
Since the late seventies, $SO_2$ vertical column densities (VCD) are provided by several ultraviolet
(UV) polar-orbiting nadir instruments, namely the Total Ozone Monitoring Spectrometer (TOMS;
Krueger, 1983), Global Ozone Monitoring Experiment (GOME; Eisinger and Burrows, 1998;
Khokhar et al., 2005), SCanning Imaging Absorption spectroMeter for Atmospheric
CHartographY (SCIAMACHY; Afe et al., 2004), Ozone Monitoring Instrument (OMI; Krotkov
et al., 2006; Yang et al., 2007, 2010; Li et al., 2013; Theys et al., 2015), Global Ozone Monitoring
Experiment-2 (GOME-2; Nowlan et al., 2011; Rix et al., 2012; Hörmann et al., 2013), Ozone
Mapping and Profiler Suite (OMPS; Yang et al., 2013, Zhang et al., 2017) and TROPOspheric
Monitoring Instrument (TROPOMI; Theys et al., 2017). From the various datasets, a remarkable
trend emerges in the ability of successive sensors to detect weaker and more localized emissions.
This is in part due to the better spatial resolution and signal-to-noise of the modern UV
spectrometers (see e.g. Fioletov et al., 2013; Theys et al., 2019), but also from advances in retrieval
techniques. In particular, the Principal Component Algorithm (PCA) applied to OMI (Li et al.,
2013, 2020a) and OMPS (Zhang et al., 2017) proved to be a very efficient method to reduce
retrieval noise and biases and thus to increase the sensitivity of the retrievals to weak $SO_2$
emissions to 30-40 kt yr$^{-1}$. This enabled major improvements in bottom-up emissions inventories
(Liu et al., 2018) and detection of missing $SO_2$ emission sources (Fioletov et al., 2016; McLinden
et al., 2016).
TROPOMI, launched in October 2017 onboard the ESA and Copernicus Sentinel-5 Precursor (S-
5P) platform, is the first atmospheric mission with a dedicated focus on the tropospheric
composition (Veefkind et al., 2012). With a spatial resolution as good as 3.5 x 5.5 km² per ground
pixel (3.5 x 7 km² before August 2019), it is specifically designed to monitor atmospheric
constituents from urban to global scales. The first observations of $SO_2$ by TROPOMI were
focusing on relatively large volcanic sources and indeed revealed the great potential of the
instrument to inform about global volcanic $SO_2$ degassing with high resolution and unprecedented
sensitivity (Theys et al., 2019; Queiβer et al. 2019). However, further investigation of
anthropogenic and volcanic $SO_2$ sources using TROPOMI revealed problems with the current
TROPOMI $SO_2$ retrievals for weak emission sources (Fioletov et al., 2020). In brief, large-scale



and variable VCD biases on the order of 0.25 Dobson Unit (DU; 1 DU= 2.69 x $10^{16}$ molecules x
$cm^{-2}$) are present in the data, which limits their use to medium to large $SO_2$ sources only.
The operational TROPOMI $SO_2$ algorithm is based on the Differential Optical Absorption
Spectroscopy technique (DOAS; Platt and Stutz, 2008), and essentially works in three steps
(details are given in Theys et al., 2017): a spectral analysis yielding $SO_2$ slant column densities
(SCD), an empirical background correction of the SCDs and a radiative transfer calculation of air
mass factors (AMF) to convert the corrected SCD into the VCD output (VCD=$SCD_{cor}$/AMF). As
a matter of fact, the $SO_2$ SCD retrieval is subject to spectral misfits which can lead to systematic
offsets. These SCD errors are difficult to correct and arise from imperfect DOAS forward
modeling. Here, we propose an alternative spectral fitting approach, named COBRA, which
strongly reduces the $SO_2$ SCD biases for the weak $SO_2$ columns and suppresses the need for the
post-processing background correction. COBRA is akin to the PCA approach, which constitutes
the basis of the OMI and OMPS $SO_2$ operational retrievals (Li et al., 2020b, 2020c). As
demonstrated below, COBRA significantly improves the quality as compared with the current
TROPOMI DOAS operational $SO_2$ product. The analysis of two and a half years of data
oversampled at high resolution reveals many new $SO_2$ emission sources globally, highlighting the
great performance of COBRA in terms of $SO_2$ detection.
The paper is structured as follows. Section 2 describes the algorithm and its practical
implementation. In section 3, $SO_2$ retrievals from COBRA are evaluated against other satellite data
sets, model results and ground-based observations. Section 4 presents long-term averaged global
results. In Section 5, we apply an emission inversion scheme to the COBRA $SO_2$ data set and
compare with previously estimated $SO_2$ emissions from the TROPOMI operational product. New
$SO_2$ emission sources detected by the COBRA are discussed. Conclusions are given in Section 6.
**2. METHODOLOGY**
**2.1 TROPOMI**
In this study, we use observations from the TROPOMI instrument on the Sentinel-5 Precursor
satellite. TROPOMI is a hyperspectral nadir sensor measuring solar radiation backscattered by the
atmosphere and reflected by the Earth, in the ultraviolet, visible, near-infrared and shortwave
infrared wavelength regions. TROPOMI delivers column amounts of minor atmospheric
constituents, such as $O_3$, $NO_2$, $SO_2$, HCHO, CO, $CH_4$, as well as aerosol and cloud information



(Veefkind et al., 2012). The S-5P satellite is a polar orbiting platform crossing the equator at
13:30h local time. A nearly global coverage is achieved in one day owing to a 2600 km wide
swath. The footprint on the ground of the satellite measurement depends mainly on the across-
track position in the swath and on the spectral band. For $SO_2$, the ultraviolet spectral band 3 is
used, and the swath is divided into 450 across-track positions (also referred to as 'rows'). The
spatial resolution for the center of the swath is approximately 3.5 x 7 km² (across-track x along-
track) until 6 August 2019 when the sampling improved to 3.5 x 5.5 km².
For this work, we analyze data measured between 1 April 2018 and 31 December 2020, and solar
zenith angles (SZA) less than 60°.

## 2.2 Algorithm description

As mentioned above, the operational TROPOMI $SO_2$ algorithm is based on the DOAS technique,
the most widely used method to derive atmospheric trace gas constituents in the UV-visible
spectral range. The inverse problem can be expressed (employing the notation of Rodgers, 2000):
$$y = K \cdot x + \epsilon \tag{1}$$
where $y = - log (I / I_o)$ is the optical depth, i.e. the logarithmic ratio of the wavelength calibrated
measured intensity $(I)$ and the reference intensity spectrum $(I_o)$ over a given wavelength range, $x$
is the state vector including SCDs of relevant trace gases and closure fit parameters (e.g. for
broadband effects), $K$ is the forward model matrix with absorption cross-sections and other spectra,
and $\epsilon$ is the measurement noise. The solution can be approximated by least-square fitting:
$$\hat{x} = (K^T S_\epsilon^{-1} K)^{-1} K^T S_\epsilon^{-1} y \tag{2}$$
where $S_\epsilon$ is the measurement error covariance matrix. The latter matrix is most often taken diagonal
(no error correlations) or proportional to unity (unweighted least-square). Eqs. (1) and (2) describe
the simplest DOAS approach and are given here for illustration purpose only. In practice, the
DOAS problem is fundamentally non-linear in many aspects and DOAS software packages, such
as QDOAS (Danckaert et al., 2017), support different non-linear retrieval options (e.g. for
wavelength shift and squeeze, or intensity offset), with the aim to improve the quality of the
retrievals.
For weakly absorbing tropospheric species, retrieval artefacts are frequent with DOAS (notably
for satellite nadir geometry), and are attributed to spectral interferences, imperfect forward model
and incomplete treatment of instrumental effects (e.g., polarization sensitivity). For UV nadir $SO_2$



retrievals in particular, biases in the data arise mainly from strong ozone absorption and imperfect
treatment of the non-elastic Rotational Raman Scattering (Ring) effect. It is generally difficult to
completely remove these offsets even after applying post-processing background corrections
(Theys et al., 2017; Fioletov et al., 2020).
The Covariance-Based Retrieval Algorithm (COBRA) presented here, and illustrated for
TROPOMI measurements, aims at correcting most of the artefacts in the DOAS $SO_2$ SCDs by
optimally retrieving a single parameter: the $SO_2$ SCD.
First introduced by von Clarmann et al. (2001), the retrieval approach was developed by Walker
et al. (2011) for nadir observations of $SO_2$ and $NH_3$ from the Infrared Atmospheric Sounding
Interferometer (IASI). Then, the technique, also known as Hyperspectral Range Index (HRI), has
been further refined and successfully applied to other trace gases and aerosols (e.g., Van Damme
et al., 2014; Franco et al., 2018; Clarisse et al., 2019). The method proved to be very sensitive and
led to superior data quality both in terms of precision and accuracy. Surprisingly, this technique
has, to our knowledge, never been applied in the UV-visible spectral range.
Starting from Eq. 1, we assume the measurement vector can be linearized around a background
$SO_2$-free spectrum $\bar{y}$:

$$y = \bar{y} + k.SCD + \epsilon_{bg} + \epsilon$$

17                                                                              (3)

with $\epsilon_{bg}$ being the uncertainty on the $SO_2$-free spectrum, and $\epsilon$ is the measurement noise. The $SO_2$
contribution to the measured spectral optical depth is approximated by the product of the
instrument slit convolved absorption cross-section vector $k$ (expressed in cm²/molecule) and the
$SO_2$ slant column density $SCD$ (in molecules/cm²). Here, we use as input of the retrieval the same
$SO_2$ absorption cross-section data (Bogumil et al., 2003) as for the operational TROPOMI $SO_2$
retrievals (Theys et al., 2017), and the wavelength interval is 310.5 – 326 nm (see discussion
below).
The basic principle of the method is to consider all contributions to the difference $(y - \bar{y})$ other
than $SO_2$ as an error term $(\epsilon_{bg} + \epsilon)$ with a Gaussian distribution. If one can define an ensemble Y
of N measured spectra, representative of the total $(\epsilon_{bg} + \epsilon)$ variability, and characterized by a mean
measurement vector $\bar{y}$ and a covariance matrix $S$:

$$S = \frac{1}{N-1}.\sum_{i=1}^{N}(y_i - \bar{y})(y_i - \bar{y})^T$$

29                                                                              (4)



then the solution of the problem writes as:

$$\widehat{SCD} = \overline{SCD} + (k^T S^{-1} k)^{-1} k^T S^{-1} (y - \bar{y}) \tag{5}$$

where $\overline{SCD}$ is the mean $SO_2$ SCD of the ensemble ($\overline{SCD} = 0$ by definition). It follows that the error
on the retrieved SCD is given by:

$$\widehat{SCD}_{err} = \sqrt{(k^T S^{-1} k)^{-1}} \tag{6}$$

Fundamentally, COBRA generalizes the measurement error covariance matrix of Eq. 2 by
incorporating geophysical background spectral variability (including all cross-correlations),
variability from the atmosphere or induced by instrumental changes.
For spectra where no enhancements of $SO_2$ can be detected, the linearization (Eq. 3) simplifies to

$$y - \bar{y} = \epsilon_{bg} + \epsilon$$

$$\tag{7}$$

Both sides of the equation have therefore the same probability distribution, and it follows that the
covariance matrix associated with $\epsilon_{bg} + \epsilon$ can readily be constructed by applying Eq. 4 on a
representative set of $SO_2$-free spectra. The key is to define the ensemble Y such that $y - \bar{y}$ cancels
much of the systematic components of $\epsilon_{bg}$.
A remarkable feature of COBRA is its simplicity. The $SO_2$ SCD retrieval in Eq. 5 reduces to a
simple dot product between the $y - \bar{y}$ residue and $k^T S^{-1}$ (skipping the normalization factor
$(k^T S^{-1} k)^{-1}$). The vector $k^T S^{-1}$ essentially contains the weights of each wavelength to the
retrieved target column amount; the strength of the method relies in the fact that these weights are
optimally determined by the measurements themselves. This is in contrast to the DOAS approach
which mostly considers all wavelengths equal. Furthermore, DOAS also allows for cross-talks
between the state vector elements, which can lead to an increase of the SCD data scatter (in
particular for weak absorbers). This is obviously not the case for COBRA, as only a single
parameter is retrieved, the $SO_2$ slant column. COBRA has other great advantages that we briefly
outline here (read also Walker et al., 2011):
-   The algorithm does not require a reference spectrum ($I_o$). Indeed, equations 4 and 5 involve
differences of logarithmic intensity ratios and thus $I_o$ cancels out. Following the same logic,
any constant spectral feature multiplicative to the radiance and shared by the ensemble Y
will have no influence on the retrieved SCDs.



- The COBRA results display low noise. This is a direct result of the COBRA approach in that the wavelengths with the largest background radiance variability will have the lowest weights on the retrieved SCD (Eq. 5).

- Very small biases are observed in the COBRA data (see next section). As a consequence, an empirical SCD background correction is not needed.

- The approach works in principle for any wavelength range. This allows flexibility in case of lower instrumental performance for certain wavelength regions.

- The covariance matrix $S$ and mean measurement vector $\bar{y}$ can be pre-calculated and the implementation of COBRA then becomes very efficient in terms of processing time (about an order of magnitude faster than DOAS non-linear schemes).

However, the practical implementation for COBRA require some caution. The main difficulty lies in the definition of the ensemble Y used to construct $S$ (and $\bar{y}$). The sample of N spectra should be highly representative of the measurement conditions under consideration, otherwise offsets in the SCDs will likely occur. Also, in principle, the spectra should be uncontaminated by absorption of the trace gas of interest. Finally, N should be large enough to insure statistically meaningful covariance results. For the retrieval of $SO_2$ from TROPOMI, we have conducted a number of tests and come to the following implementation choices.

The input spectra for the covariance matrix calculation are analyzed separately for each TROPOMI row. We also treat each orbit individually to account best for the orbit-to-orbit variability. The data are first screened for solar zenith angles larger than 60°, and to cope with the latitudinal dependence, the data are divided into 6 equal and non-overlapping along-track segments. For each segment, an initial covariance matrix $S$ is derived and initial estimates of $SO_2$ SCDs are inverted through equation 5. In a second step, improved estimates of $S$ and $SO_2$ SCDs are obtained iteratively by removing $SO_2$ contaminated spectra from the ensemble Y. To do this, we use the ratio of the $SO_2$ SCD to its retrieval uncertainty (Eqs. 5 and 6), referred to as the signal to noise ratio (SNR):

$$SNR = \frac{k^T S^{-1}.(y-\bar{y})}{\sqrt{k^T S^{-1} k}} \tag{8}$$

A fixed SNR upper value of 1.5 is used for the filtering and the number of iterations is set to 4. A lower limit on the number N of $SO_2$-free spectra is set to 50. If this limit is reached, because of a





major volcanic eruption for example, the SO₂ SCD retrieval is entirely skipped for the
corresponding row-segment pair.
It should be stressed that COBRA is close in concept to the PCA SO₂ algorithm of Li et al. (2013,
2020a). In brief, the PCA scheme characterizes the background radiance variability using a number
of leading PC spectra (typically 20-30), instead of a covariance matrix. The SO₂ column is then
retrieved from the measured spectrum along with the PCs fitted parameters. In comparison,
COBRA removes the need of having many parameters to fit. Only the SO₂ slant column density is
determined and the background radiance variability is fully described by the covariance matrix. In
a sense, COBRA can be considered as a generalization of the PCA scheme. It is therefore of great
interest to compare the two methods (see section 3.1). Having this perspective in mind, we have
made a number of choices to facilitate the comparison. For instance, we have used a spectral
window from 310.5 to 326 nm (instead of 312-326 nm for the TROPOMI operational DOAS
product), which includes the same strong SO₂ absorption bands as in the spectral range 310.5-340
nm used by Li et al. (2013). This choice is also motivated by the inclusion of the intense absorption
band at 310.8 nm which leads to a further reduction of the noise on the SO₂ column by about 25%.
Note that initial tests with the TROPOMI operational algorithm using the 310.5-326 nm window
were actually not very successful (large SO₂ SCD offsets). On the contrary, with COBRA, we
tested both wavelength ranges (310.5-326 nm and 312-326 nm) and found only small differences
between the retrieved SO₂ column patterns (Fig. S1).
In the following sections, SO₂ vertical columns will be presented. For the SCD to VCD conversion,
we have used air mass factors from the operational product. Note that doing so is not strictly valid
because one should expect lower AMFs due to the change in fitting window (from 310.5 to 326
nm to 312-326 nm). To account for this, we have applied a constant scaling factor of 1.15 to the
retrieved SO₂ VCDs. Based on radiative transfer calculations, we found this to be a good first order
correction. However, in the future, AMFs shall be recalculated properly. For the cloud filtering
and AMF cloud correction, the operational cloud product OCRA/ROCINN CRB is used (Loyola
et al., 2018; Compernolle et al., 2020).
As a final note, it should be reminded that the operational TROPOMI algorithm also handles the
retrieval of large SO₂ VCDs, by making use of multiple fitting windows (as described in Theys et
al., 2017). In this study, we have not applied COBRA on the alternative fitting windows. While





there is no fundamental limitation to do so, COBRA is relevant mostly for low $SO_2$ columns. All
the results presented in the next sections are for situations where the $SO_2$ VCDs are below 5 DU.
## 3. VERIFICATION OF THE RETRIEVALS
### 3.1 Comparison to satellite observations and CAMS
In order to evaluate the $SO_2$ data from COBRA, it is interesting to first investigate the bias and
data scatter over a clean region and compare with the operational product (hereafter referred to as
'DOAS'). In Figure 1, the mean and standard deviation of $SO_2$ slant columns over an equatorial
Pacific region are shown for one particular orbit, as a function of the TROPOMI row. As can be
seen from Fig. 1a, the DOAS data suffer from SCD offsets in the range of $\pm 0.25$ DU, despite the
background correction applied. These offsets have a low-frequency dependence component with
the across-track position but also vary sharply from one row to the next (leading to stripes in the
$SO_2$ maps). In contrast, the COBRA results have very small SCD biases (mostly below $\pm 0.025$
DU) and no noticeable across-track dependence. It follows that COBRA is a very powerful bias
self-correction and destriping scheme. In Fig. 1b, the standard deviations of the $SO_2$ SCD values
are shown for both algorithms. Compared to DOAS, it is clear that the data scatter is significantly
improved with COBRA, by a factor of 2. It is understood that part of this noise reduction is due to
the change in fitting window (section 2.2), but most of the improvement (~75%) is from the
COBRA approach. From Fig. 1, it is clear that the combined reduction of bias and data scatter
provided by COBRA over the DOAS results is very significant. From a practical point of view, a
factor of 2 improvement of the data scatter means 4 times less pixels to average to reach a certain
noise level.
In Fig. 1b, we note also a distinct increase in data scatter for the outermost rows, for both DOAS
and COBRA. This feature is due to difference in detector signal binning at the swath edges which
leads to an increase in radiance shot noise. To keep the data of the best quality, we will not use the
50 outermost rows in the following of the paper.



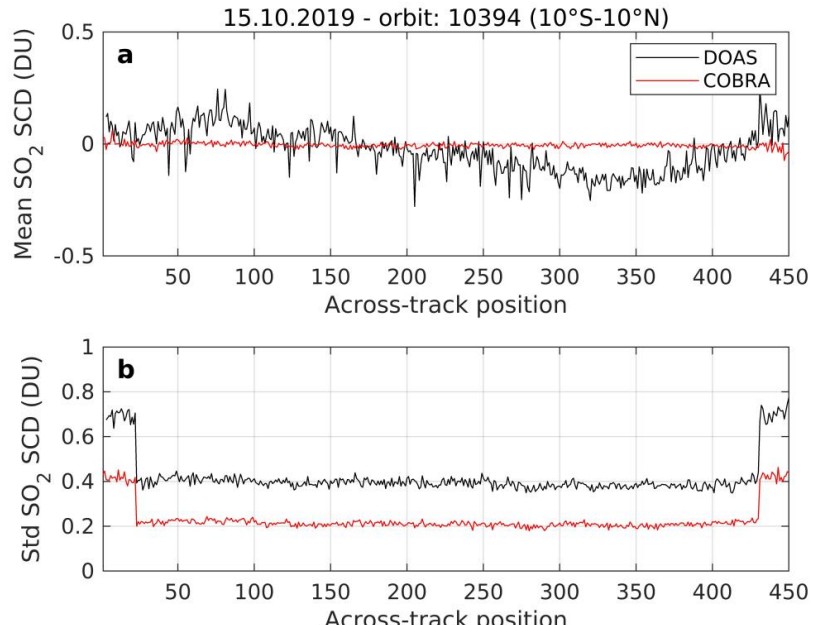

Figure 1. (a) Mean $SO_2$ slant columns from (black) DOAS (background corrected) and (red)
COBRA for one orbit (10394 on 15 October 2019) over the equatorial Pacific region (10°S-10°N),
as a function of the across-track position of TROPOMI, (b) same as (a) for the $SO_2$ SCD standard
deviation.
Figure 2 compares the DOAS and COBRA seasonal averaged $SO_2$ VCD maps from September to
November 2019. The data are gridded at a resolution of 0.1° x 0.1° and smoothed by a 2-
dimensional 5-points box car function. Both DOAS and COBRA results are extracted using
identical pixel selection criteria: SZA less than 60°, radiometric cloud fraction lower than 30% and
TROPOMI rows 26-424. From Fig. 2, several artefacts are evident in the DOAS product. Negative
values are found in the tropics and a large scale positive bias at mid-latitudes. In comparison,
COBRA remarkably solves all the systematic biases found in the operational product whereas the
signal from major $SO_2$ sources (e.g. in China, India, Middle-East, South Africa, Central and South
America) is nicely preserved. Note that for individual pixels with unambiguous detection of $SO_2$
(typically $SO_2$ VCDs larger than 2 DU), the agreement between DOAS and COBRA is excellent
(see e.g., Fig. S2). In Fig. 2, a closer look at the COBRA $SO_2$ map still reveals some negative
values for specific locations. For instance, the Garabogazköl Basin near the Caspian Sea is



particularly visible. It is characterized by a salt flat with a high albedo. This surface effect is
apparently poorly represented in the radiance covariance, and leads to the negative values observed
in the data.
Retrieval results using the new COBRA are also evaluated in Fig. 2 against a scientific TROPOMI
$SO_2$ product generated using the PCA approach. The settings of the experimental TROPOMI PCA
$SO_2$ algorithm, including the spectral range and number of iterations, are identical to the
operational OMI algorithm with the following exceptions: 1) TROPOMI pixels from each row are
grouped into sectors of 20-degree latitude bands, instead of three sectors as in the OMI algorithm;
2) a third degree polynomial is removed from each Sun-normalized radiance spectrum before PCA
analysis; 3) at maximum 20 PCs are used in the fitting instead of 30 in the current OMI algorithm
(Li et al., 2020a); and 4) no attempts were made to reduce TROPOMI retrieval noise over the SAA
affected areas. For this exercise, the PCA scheme uses as input the same $SO_2$ absorption cross-
section data (Bogumil et al., 2003) as for the DOAS and COBRA retrievals, and the same selection
of pixels. Figure 2 also compares the TROPOMI $SO_2$ columns (from DOAS, COBRA and PCA)
to the operational OMPS $SO_2$ PCA retrievals NMSO2_PCA_L2 V2 (Zhang et al., 2017; Li et al.,
2020c). Although OMPS has a coarser resolution (50 x 50 km²) than TROPOMI, it provides
nonetheless a useful reference data because it operates on the Suomi National Polar-orbiting
Partnership (SNPP) satellite which flies in loose formation with S-5P (i.e. 3-5 minutes difference
of overpass time). To allow a meaningful comparison, the OMPS pixels were selected similarly as
TROPOMI, i.e. with cloud radiance fraction lower than 30% and OMPS across-track positions 3-
34. Note finally that to avoid discrepancies due to different a-priori profiles in the TROPOMI and
OMPS retrievals, a fixed AMF of 0.4 was used for all four data sets. As can be seen from Fig.2,
an overall excellent agreement is found between COBRA and PCA retrievals, the observed $SO_2$
spatial distributions being essentially the same. However, the OMPS $SO_2$ data set has different
patterns over China (possibly due to sampling differences), and also appears noisier than the
TROPOMI results (as expected from the smaller number of pixels). When comparing the
TROPOMI COBRA and PCA maps, very consistent results are found. Yet, the quality of COBRA
seems slightly better than the PCA retrievals. In particular, COBRA is much less sensitive to the
South Atlantic Anomaly than PCA data, which exhibit many outliers in the corresponding region.
At mid-latitudes, there is also a slight positive bias (of about +0.1 DU on average) and higher noise
in the PCA results compared to COBRA.

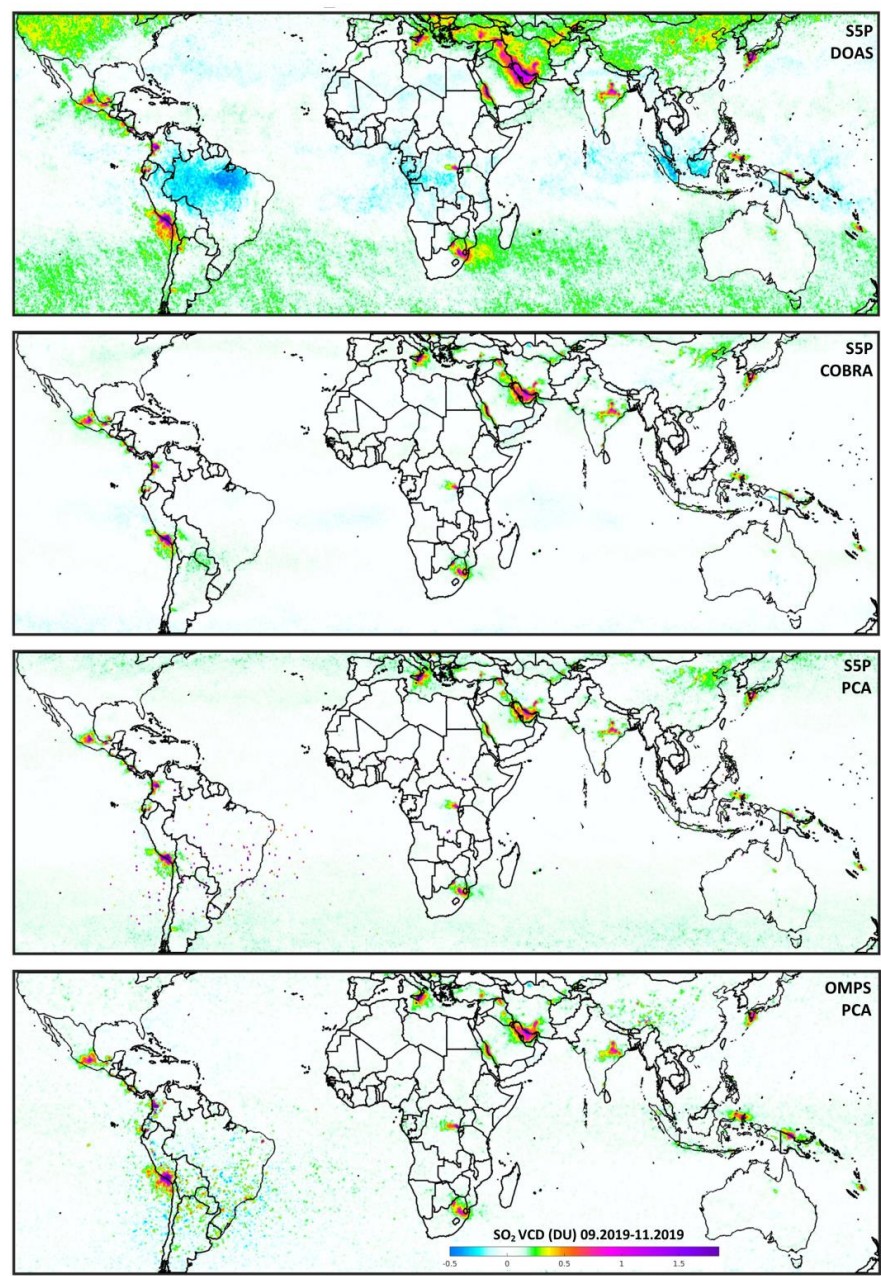

Figure 2. Comparison of seasonal mean $SO_2$ columns for September to November 2019 retrieved from TROPOMI DOAS, COBRA, PCA and OMPS PCA algorithms (from top to bottom). Consistent pixel selection criteria, gridding and retrieval settings are applied (see text). For all four data sets, a fixed AMF of 0.4 is applied.



We have estimated the data scatter for the three TROPOMI data sets, based on measurements from
the same orbit over the Pacific as Fig. 1. Results are shown in Figure 3, as a function of latitude.
We find that COBRA has a SCD noise level 20-25% lower than the PCA retrievals, and twice
better than DOAS (as in Fig. 1). Translating the numbers of Figure 3 in terms of vertical columns
for a typical pollution scenario, we estimate the retrieval precision for individual pixels typically
to be 0.5 - 1 DU for COBRA.

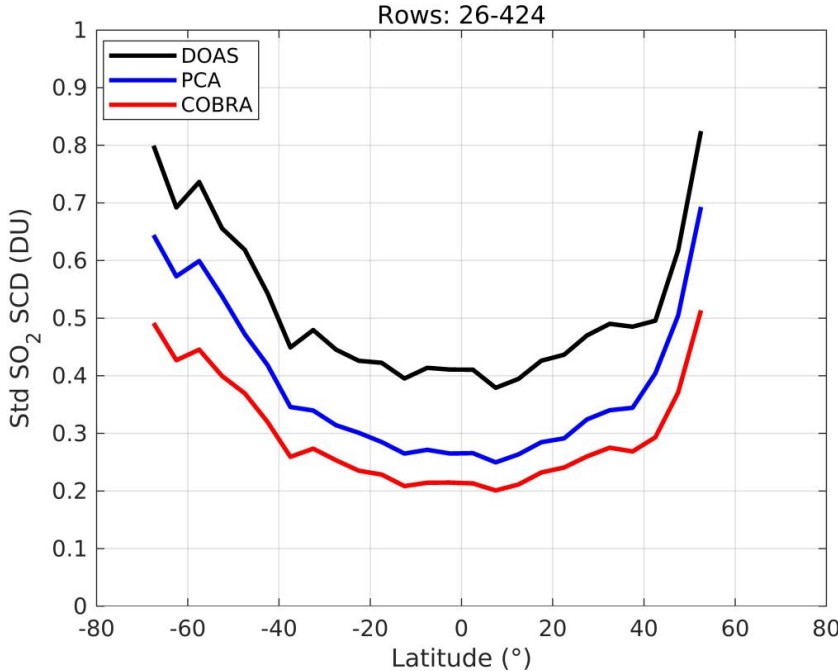

Figure 3. Standard deviation of the $SO_2$ slant columns as retrieved from DOAS (black), PCA (blue)
and COBRA (red) for one TROPOMI orbit (10394 on 15 October 2019, same as Fig. 1) for rows
26-424, as a function of latitude (for 5° bins).
To further evaluate the overall quality of the COBRA retrievals, the $SO_2$ VCDs can also be
compared to model data. Here, we have used the output of the Copernicus Atmosphere Monitoring
Service (CAMS; https://atmosphere.copernicus.eu/) regional model, for September to November
2019. The CAMS regional air quality production is based on an ensemble of 9 European air quality
models that are run at a resolution of 0.1° and produce 4-day, daily forecasts of the main
atmospheric pollutants, including $SO_2$. The forecasts and analyses from all 9 models are combined



in calculating the median value of the individual outputs, which is designated as the ENSEMBLE
output and is the field used in this study. The CAMS regional ensemble data was obtained from
the Copernicus Atmosphere Data Store (ADS, https://atmosphere.copernicus.eu/data). More
information about the CAMS regional system can be found on the ECMWF website
(https://confluence.ecmwf.int/display/CKB/CAMS+Regional:+European+air+quality+analysis+a
nd+forecast+data+documentation). The CAMS regional system used the CAMS-REG-
AP_v2_2_1 emissions (reference year: 2015) between June 2019 and February 2020, and the
updated CAMS-REG-AP_v3_1 emissions dataset (reference year: 2016) since February 2020.
In Figure 4, seasonal regional maps of S-5P $SO_2$ VCDs over Eastern Europe from the DOAS and
COBRA schemes are compared to the output of the CAMS regional model, for September to
November 2019. From the maps, it is clear that the COBRA results are in much better agreement
with the CAMS analysis than the DOAS data. Owing to the quasi-absence of bias and the low
noise level, the COBRA data allows better isolation of the emission sources. The agreement
between COBRA and CAMS is however not perfect and there are several explanations for this.
Most of the $SO_2$ emissions in this region are from coal-fired power plants and the emission
inventory used by CAMS is likely not reflecting neither the actual activity nor the emission
mitigation solution (e.g. $SO_2$ scrubbers) at each power plant. Noteworthy is also the absence of
$SO_2$ emissions from Mt. Etna in CAMS. Secondly, the AMFs used here are calculated with $SO_2$
profiles from TM5, a different model with a coarser resolution (1° x 1°) than CAMS regional.
Therefore the COBRA and CAMS $SO_2$ columns cannot be strictly compared. Nevertheless, the
comparison in Fig. 4 is encouraging. In the future, the COBRA $SO_2$ retrievals together with the
corresponding column averaging kernels (Eskes and Boersma, 2003) could be ingested in the
CAMS assimilation system to better constrain the model $SO_2$ output and emission estimates.

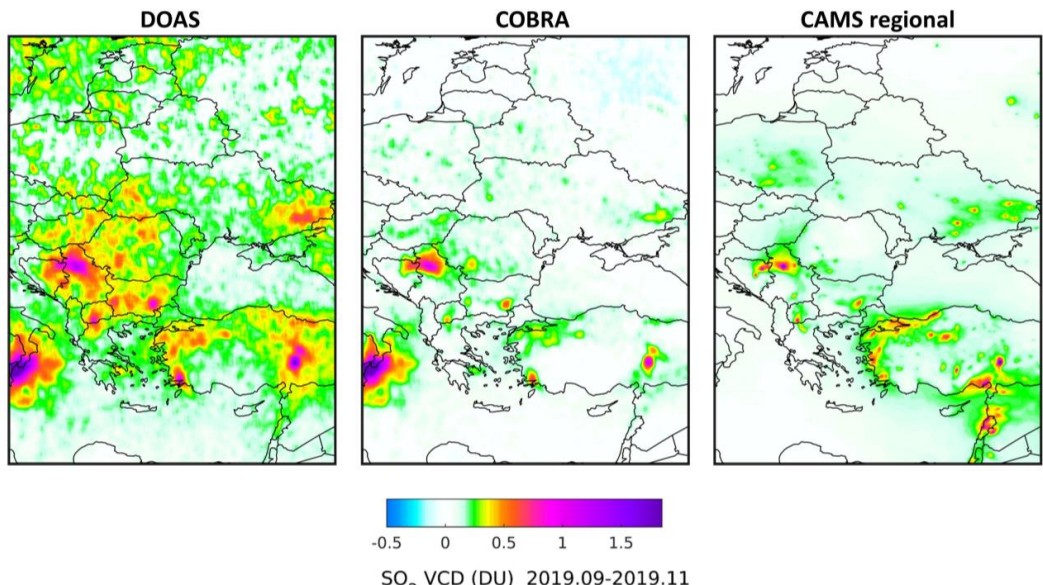

Figure 4. Seasonal mean SO$_2$ columns for September to November 2019 from (left-center) TROPOMI DOAS and COBRA retrievals, and (right) simulated by the CAMS regional model. The CAMS data are displayed at the 0.1° x 0.1° native resolution.

## 3.2 Comparison to ground-based MAX-DOAS observations

The Multi-Axis DOAS (MAX-DOAS) measurement technique is an established method to retrieve tropospheric trace gas columns and vertical profiles from a sequence of spectral observations performed at various elevation angles above the horizon (Hönninger and Platt, 2002; Tirpitz et al., 2021). MAX-DOAS measurements leverage the fact that low elevation measurements have enhanced sensitivity to atmospheric pollutants in the boundary layer and that the combination of the different elevations carries information on the vertical distribution of the trace gas of interest as well as aerosols. The simplest estimation of the tropospheric VCD from MAX-DOAS measurements is obtained by scaling the differential SCD at a given elevation angle (often 15° or 30°) with an AMF assuming a geometrical light path through the trace gas layer. Recently, more sophisticated approaches have been developed to retrieve the concentration profile in the troposphere using multiple elevation measurements.

Here we compare our TROPOMI SO$_2$ VCD data to MAX-DOAS observations at two sites, both characterized by relatively low SO$_2$ columns: Xianghe and Mohali (Table 1). In general, the





different MAX-DOAS instruments and SO$_2$ retrievals share common characteristics, practices and
approaches, and the reader is referred to the publications listed in Table 1 for more detailed
information.
For the comparison, we have used a common set of selection criteria for the satellite data. For each
day, we selected the TROPOMI pixels within a 25 km radius circle around the station of interest,
a strict radiometric cloud fraction threshold of 20%, SZA lower than 60° and AMF larger than 0.2.
If the number of retained pixels is larger than 10 then the mean SO$_2$ VCD is calculated and
compared to the averaged SO$_2$ column for the MAX-DOAS measurements within ±1 hour of the
S-5P overpass time.
Regarding the ground-based retrievals, the SO$_2$ VCDs are estimated, for Mohali, using the 15°
elevation SO$_2$ SCDs and geometrical AMFs. Conversely, the retrieved data for Xianghe consist of
SO$_2$ profiles. These are integrated along the vertical to provide the VCDs. Moreover, to make the
comparison between MAX-DOAS and TROPOMI more consistent, we have rescaled the
TROPOMI VCDs using the satellite averaging kernels (Eskes and Boersma, 2003) and the MAX-
DOAS SO$_2$ profiles at Xianghe.
Table 1. Summary of SO$_2$ VCDs comparison

| Station | Reference | MAX-DOAS VCD calculation | Period | Mean SO$_2$ VCD (DU) for SZA<50° (*50°<SZA<60°*) | | |
|---|---|---|---|---|---|---|
| | | | | **MAX-DOAS** | **S-5P COBRA** | **S-5P DOAS** |
| Xianghe, China 39.77°N 117°E | Wang et al. (2014) | Integrated profile | 01.2020-10.2020 | 0.2 (*0.28*) | 0.21 (*0.34*) | 0.34 (*0.77*) |
| Mohali, India 30.67°N 76.74°E | Kumar et al. (2020) | 15° elevation (geometrical) | 05.2019-10.2020 | 0.21 (*0.18*) | 0.18 (*0.26*) | 0.16 (*0.50*) |

The comparison results between COBRA and MAX-DOAS measurements are shown in Figure 5a
and 5b, for Xianghe and Mohali, respectively. Overall, the agreement between COBRA and MAX-
DOAS data is very good, keeping in mind that the levels of SO$_2$ columns are quite low. The slopes
of the regression lines are close to unity. In Table 1, the mean SO$_2$ columns from MAX-DOAS,
TROPOMI COBRA and DOAS retrievals are given at each station and for different SZAs. A
striking feature of the comparison is that the COBRA results show similar good agreements over



a wide range of SZA. It further supports the idea that COBRA yields unbiased results over varying
observation conditions. This is in contrast to the DOAS product which is clearly biased high for
high SZA.

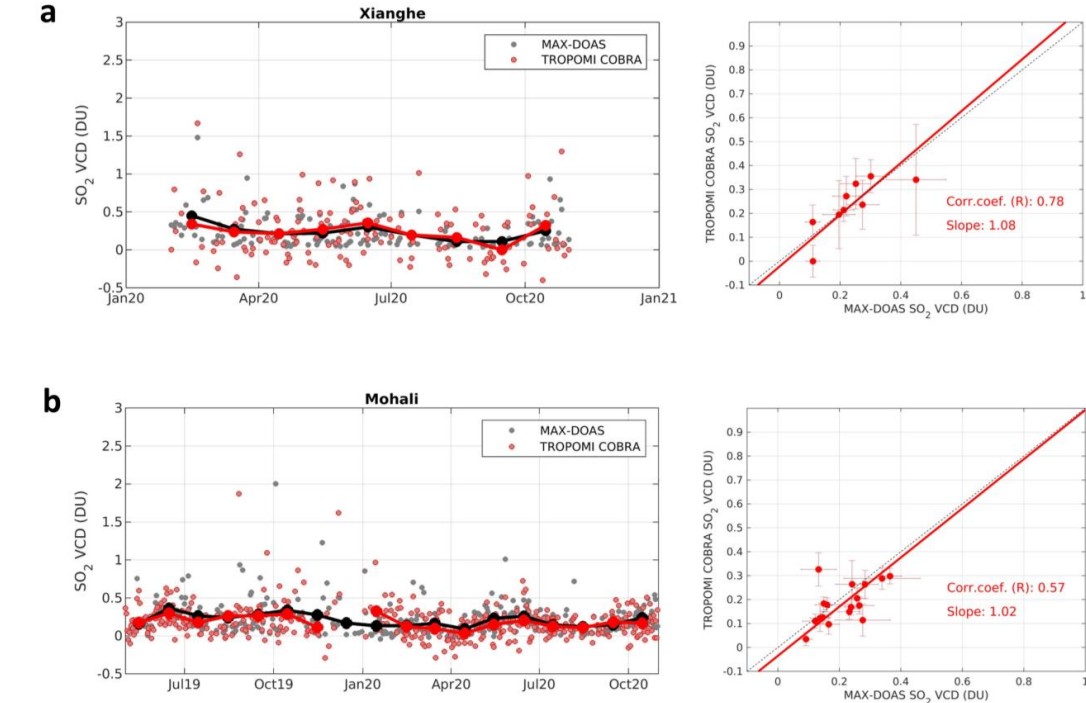

Figure 5. (left) Comparison of monthly mean $SO_2$ columns from MAX-DOAS and TROPOMI
COBRA for (a) Xianghe, and (b) Mohali. The grey and pale red dots correspond to the individual
days. (right) Scatter plots of monthly mean $SO_2$ columns of TROPOMI COBRA vs MAX-DOAS
observations. Error bars are the standard errors on the monthly average $SO_2$ columns. The
correlation coefficient and slope of the regression line are given as an inset for each plot.
**4. GLOBAL RESULTS**
In this section, we present long-term global results from COBRA, based on two and a half years
(April 2018 – December 2020) of cloud-free TROPOMI data (radiometric cloud fraction less than
30%). Using an oversampling technique, a global average $SO_2$ column map at 0.015° x 0.015°
resolution was obtained and smoothed by a 2-dimensional 10-points box car function. Figure 6
shows the resulting $SO_2$ distribution for specific regions, over East China, India, the Middle East,

South America, South Africa, US and Europe (the global map is also available in the form of a Google Earth /geotiff file, as supplementary material). Figure 6 also shows the locations of the $SO_2$ sources based on the latest OMI 2005-2019 catalogue (Fioletov et al., 2016), with a total of 588 sites, including power plants, smelters, oil and gas industry sources, and volcanoes. As can be seen, many of the sources of the catalogue are easily identified as $SO_2$ hotspots on the map. Conversely, there is also a significant number of sources in the inventory with no detectable $SO_2$ in the TROPOMI data, but one should keep in mind that the catalogue gathers emission sources since the beginning of the OMI data record in 2005 and several of these sources have ceased operations or decreased drastically their emissions since then (e.g. due to the operation of $SO_2$ scrubbers in coal power plants). To help identifying those sources, Figure 6 shows with a different marker the sources detected by OMI for 2018-2019 (i.e. the sources with emissions above the detection limit, as in Fioletov et al. (2016) and McLinden et al. (2016); see also Section 5.2). This is also helpful to highlight the differences in sensitivity, as many sources are detected by TROPOMI but not by OMI.

Generally speaking, the $SO_2$ maps of Figure 6 are very detailed. Biases over clean regions are remarkably low, and emissions-related patterns with $SO_2$ VCDs less than 0.25 DU are clearly visible in many places. By scrutinizing the $SO_2$ distributions, one can identify numerous sources from the current catalogue but also several potentially new source regions. However, some care must be taken in attributing new sources and relate this to the improved sensitivity of TROPOMI COBRA. First, the catalogue is arguably not resolving well the individual sources for dense regions (e.g., in East China and India) and, as a matter of fact, typically reports a total emission estimate of a point-like source for what is in reality a cluster of sources. While the algorithm to handle such clusters of sources exists (Fioletov et al., 2017), it has not been implemented in the catalogue yet. Second, the $SO_2$ catalogue is being populated on a best-effort basis, and a number of emission sources might be missing, in particular for emerging countries where industrial infrastructures are built quickly. Third, $SO_2$ outflow from the strong sources - or clusters of sources - can lead to variations in the map and thus fictitious emission sources. Finally, retrieval artefacts, measurement noise or sampling related issues can also lead to false sources identification. Note that a comprehensive identification and classification of new sources from the COBRA $SO_2$ data is not the scope of the present study. Here, we aim at discussing plausible new $SO_2$ sources (i.e.





not in the OMI catalogue). In Section 5.2, we will further demonstrate the excellent performance
of COBRA to detect very weak emissions, for a limited number of sources.
The new identified sources are characterized by low $SO_2$ column levels in the range 0.05 - 0.2 DU.
For instance, in Fig. 6a we observe hotspots of $SO_2$ from power plants (mostly coal but also likely
gas) in North- and South-Korea, northern Vietnam (near Haiphong), several Chinese provinces
(e.g., Hubei, Guangxi, Guangdong) and along the coast of China. In Fig. 6b, several weak emission
sources can be isolated in India (e.g., over the western coast and the Indo-Gangetic plain), Pakistan,
Bangladesh and Sri Lanka (near the city of Colombo). In the Middle East (Fig. 6c), most of the
$SO_2$ emissions are from oil and gas related industries, like power plants, gas flaring and refineries.
Examples of weak $SO_2$ emissions can be found in Saudi Arabia, Oman, Egypt, Syria (near the city
of Damascus) and Iran. In South America (Fig. 6d), new emission sources are popping up, notably,
in Brazil (near Rio de Janeiro, São Paulo and Porto Alegre), and on both sides of the Andes (in
Chile and Argentina). In South Africa (Fig. 6e), in addition to the strong emissions from the coal
power plants of the Highveld, a clear $SO_2$ signal is detected over Cape town. Interestingly, the
measured $SO_2$ distribution nicely matches the orography setting. In the US (Fig. 6f), the most
striking emission region is the state of California, with enhanced $SO_2$ over the Central Valley and
the city of Los Angeles. Over the central and eastern parts of the US, the emissions from power
plants have declined dramatically over the last 15 years (Krotkov et al., 2016). However, the data
still show enhanced $SO_2$ over some of them. In Europe (Fig. 6g), most of the observed enhanced
$SO_2$ correspond to sources already in the catalogue. Still, a number of small spots are found e.g.,
in eastern Europe (Romania, Serbia, Kosovo, Hungary), Germany (near Leipzig), Turkey and
Tunisia (Gulf of Gabes). Interestingly, enhanced $SO_2$ is also observed over the Gibraltar strait and
Red sea which might result from shipping emissions.
Overall, the $SO_2$ maps of Fig. 6 nicely illustrate the great ability of TROPOMI to detect weak $SO_2$
point emissions sources when analyzed using a sensitive approach as COBRA. Using Google Earth
imagery and information on industrial facilities location, we were able to confirm that many
features in the $SO_2$ map are real sources. For this, we have also compared our $SO_2$ data to
tropospheric $NO_2$ column maps from TROPOMI. An example of comparison is shown in Fig. S3
for a region over Central Asia. There, the $SO_2$ emissions sources in the catalogue are mostly from
coal power plants and smelters, in the Xinjiang province (China) and east Kazakhstan. As can be



seen in Fig. S3, several other $SO_2$ emission hotspots are detected (notably in the Xinjiang province) which clearly coincide with locations with enhanced tropospheric $NO_2$.

Nevertheless, several patterns in the $SO_2$ map (Fig. 6) are hard to relate to point source emissions. In particular, the $SO_2$ signal observed over Cape town (Fig. 6e) and Los Angeles (Fig. 6f) could be due to area sources rather than point emissions. Over South America (Fig. 6d) and eastern US (Fig. 6f), the apparent $SO_2$ background is intriguing. It is unclear whether this could be due to real $SO_2$ emissions or not. We also identify several artefacts in the data. Unsurprisingly, biases in the data occur for specific conditions which are under-sampled or not optimally represented by the covariance matrix. These are most often surface-related effects (due to peculiar albedo or elevated terrain). One illustration of this problem is given in Fig. 6c, over the Nile Valley. Although some real $SO_2$ emissions are found in the area, with $SO_2$ VCDs larger than ~0.1 DU, there are also unexpected enhancements in the $SO_2$ column that follow the Nile River. These are probably due to the very dark surfaces there. Similarly, elevated values are also found further South in Sudan and Ethiopia, over vegetated scenes. However, the resulting $SO_2$ VCD biases are overall very small, typically less than 0.04 DU ($\sim 1 \times 10^{15}$ molecules/cm²), and can be suppressed by a local bias correction or more sophisticated approaches.

As mentioned above, the attribution of new sources based on $SO_2$ maps is not straightforward. Efficient space-based techniques do exist, to isolate sources and estimate their emissions (Fioletov et al., 2015; McLinden et al., 2016; Clarisse et al., 2019). However, applying such methods systematically to the TROPOMI COBRA $SO_2$ data goes beyond the scope of this paper. Instead, in the next section, we will estimate the $SO_2$ emissions for the known largest sources, and demonstrate the potential of COBRA for the retrieval of weak emissions, for a limited number of new sites.



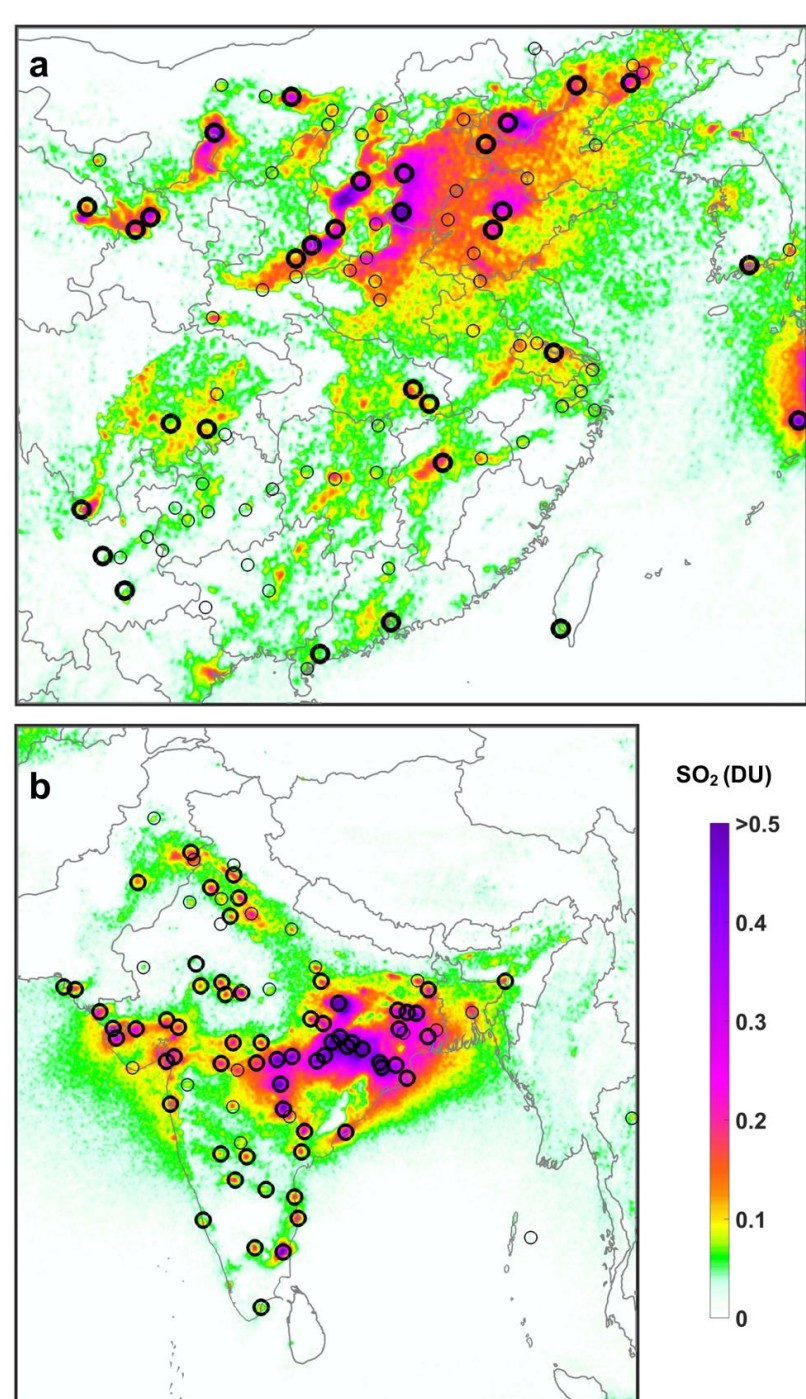





Figure 6. Averaged SO$_2$ column (in DU) for April 2018 to December 2020 over (a) East China, (b)
India, (c) the Middle East, (d) South America, (e) South Africa, (f) US and (g) Europe. The black
circles mark the locations of SO$_2$ sources detected by OMI (in bold for the 2018-2019 period, see
text). Due to the massive eruption of Raikoke on 21 June 2019, all data in the northern hemisphere,
for the 3 months period after the eruption, are filtered out.

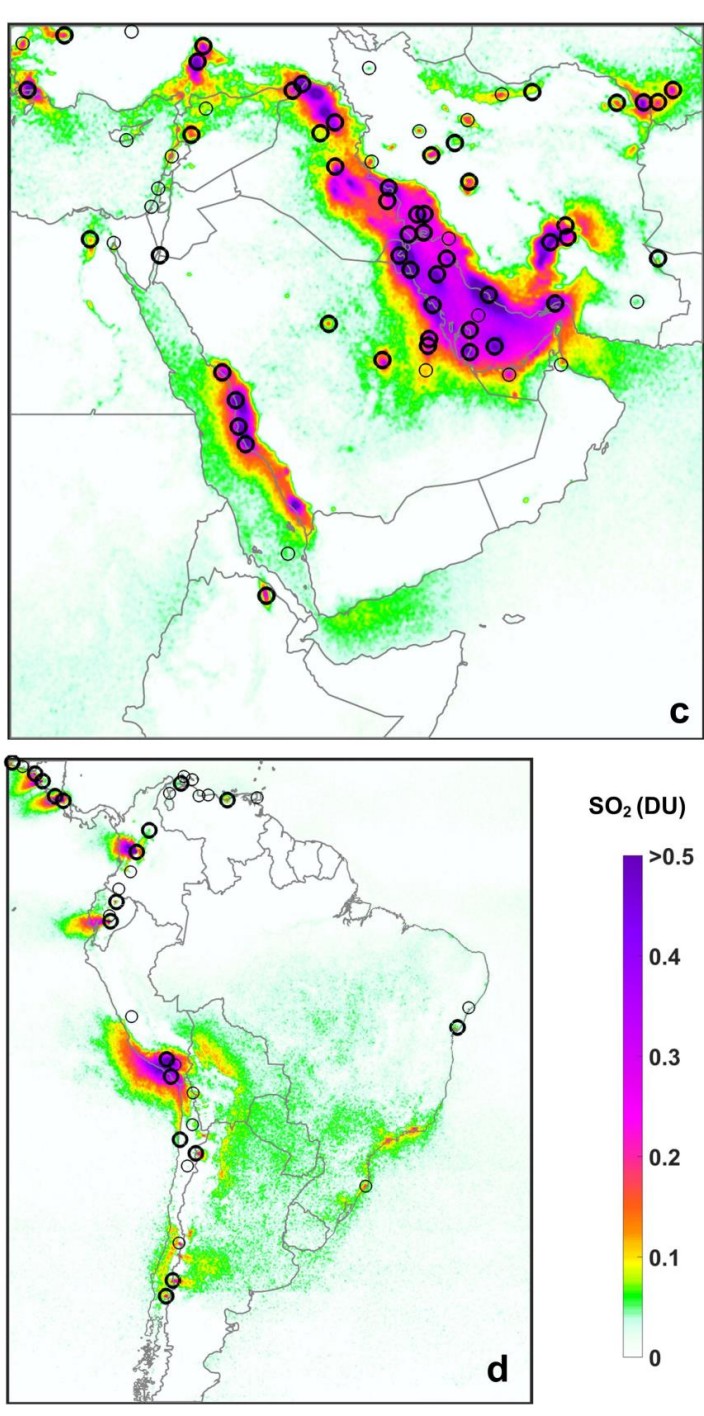

2    Figure 6. Continued.



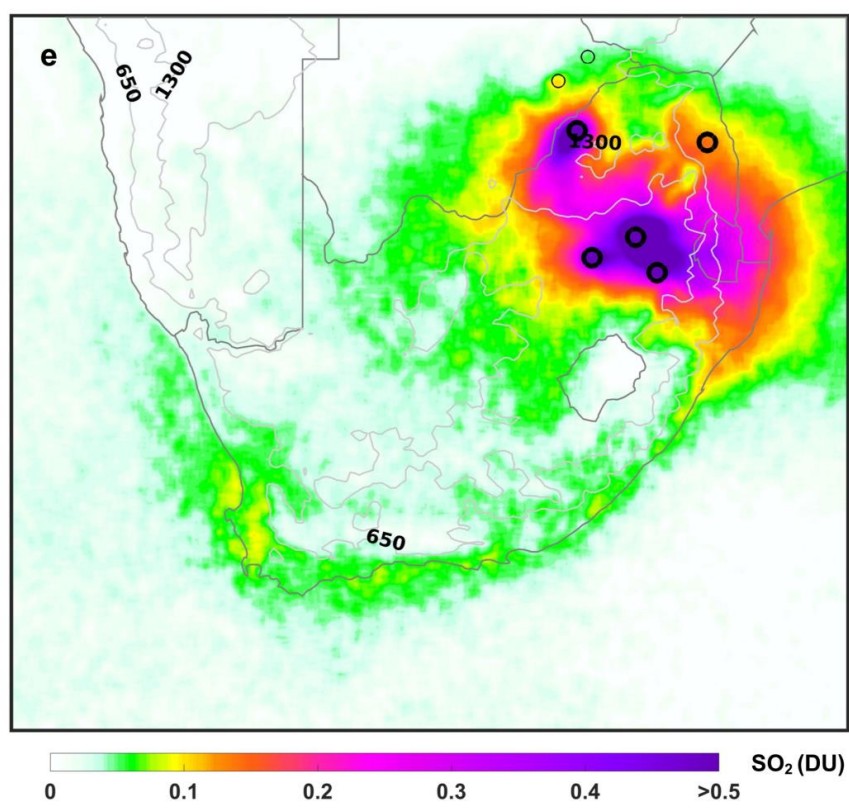

Figure 6. Continued. The gray lines are the topography isolines (in meter). Note that to further
reduce the data scatter, the $SO_2$ map was smoothed by a 2-dimensional 20-points box car function
(instead of 10-points function for the other sub-figures).



SO$_2$ (DU)

0    0.1    0.2    0.3    0.4    >0.5

2
3    Figure 6. Continued.





## 5. EMISSIONS ESTIMATES

Satellite observations are being increasingly used to estimate $SO_2$ emissions. In particular, new methods have been very successful in deriving reliable emission rates, and even detecting missing sources, by combining satellite $SO_2$ columns and wind information, without the need of atmospheric chemistry transport models (e.g., Beirle et al., 2014; Fioletov et al., 2016, 2017, 2020; McLinden et al., 2016; Carn et al., 2017). These techniques have been used to derive global $SO_2$ emissions inventory from OMI observations (Liu et al., 2018). Recently, Fioletov et al. (2020) presented an analysis using the TROPOMI operational $SO_2$ product and found overall consistent results with the OMI emissions estimates. The TROPOMI-based emissions uncertainties were found a factor of 1.5 - 2 lower than the ones from OMI. In this section, we repeat the same analysis using the COBRA $SO_2$ retrievals and investigate the added value of COBRA for the estimation of $SO_2$ emissions. The details of the inversion technique can be found in Fioletov et al. (2015) and references above.

In brief, the method considers a potential point source and apply a wind rotation of the satellite measured $SO_2$ VCDs around this location. This first step enables to align all plume dispersion patterns along a fixed direction and leads to an improved $SO_2$ detection limit. By contrasting the upwind and downwind averaged $SO_2$ columns, the wind rotation procedure allows to confirm whether the test location is a real emission source and also to correct for a possible bias in the data. Note that for this first step, the retrieved $SO_2$ VCDs are rescaled using site-specific AMFs so that realistic $SO_2$ emission profile shapes (based on the elevation of the site and climatological boundary-layer height) are used for all analyzed sources.

The second part of the retrieval method deals with the emission estimate itself. The averaged downwind $SO_2$ field is modelled by an exponential modified Gaussian function which accounts for the $SO_2$ total mass, e-folding time and plume width. From the fitted parameters, the average $SO_2$ emission rate can be derived directly. Here the baseline inversion is however not to fit all three parameters but rather to prescribe the e-folding time and plume width, and therefore the only parameter derived from the fit is the $SO_2$ total mass which is directly proportional to the $SO_2$ emission rate.



**5.1 SO$_2$ emissions for large sources**

The method was applied to the SO$_2$ data from COBRA for 274 large emissions sources, including
power plants, volcanoes, oil and gas sources and smelters, distributed worldwide. In Fig. 7a, the
results are compared to the analysis of Fioletov et al. (2020) using the TROPOMI DOAS product,
for the period from April 2018 to March 2019.

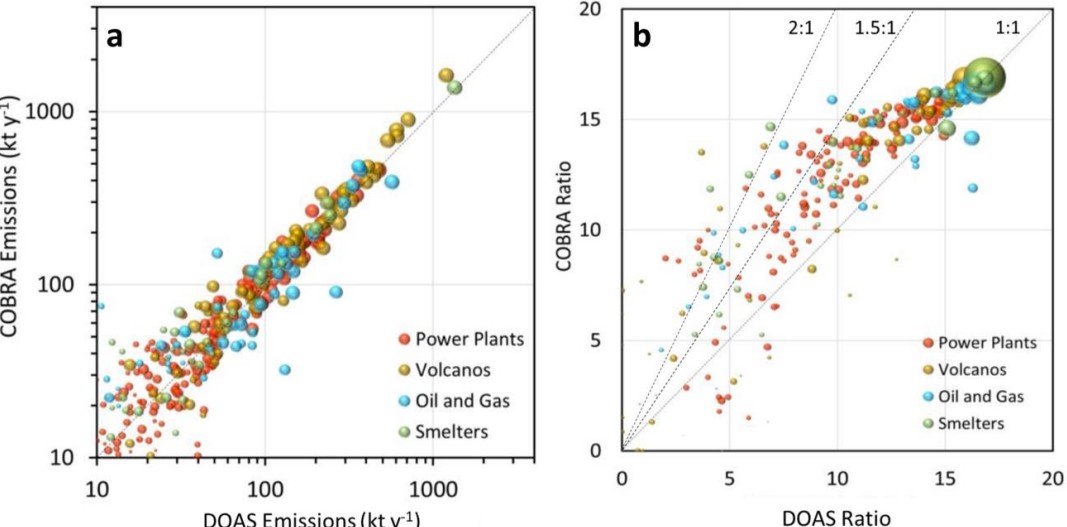

Figure 7. (a) Estimated SO$_2$ emissions from TROPOMI, based on the COBRA and DOAS
algorithms analyzed for power plants, volcanoes, oil and gas industries, smelters sources. (b)
Ratios of the estimated emissions and the corresponding uncertainties. The size of the marker is
proportional to the estimated SO$_2$ emission.
In general, the emission estimates from COBRA and DOAS are fairly consistent for all four source
types. However, it was found that the local bias in the satellite data (as derived from the upwind
SO$_2$ columns) are much lower with COBRA (~ 0.05 DU) than with DOAS (~0.25 DU), consistent
with previous findings (Section 3). Also, as result of the large improvement in the noise level, the
estimated emissions uncertainties are significantly improved with COBRA compared to DOAS,
by 20-50% on average (see Fig. 7b).
It should be emphasized though that the improvement of emission uncertainties depends on the
emission level. The sources considered here are relatively large sources that have been previously



detected by OMI. The TROPOMI COBRA $SO_2$ data set presented in this study combines the
advantages of high spatial resolution, low noise level and almost no bias. It has therefore the
potential to detect weaker sources (as shown in Sect. 4).
**5.2. Detection of weak emissions**
It is enlightening to estimate the lowest level of $SO_2$ emission detectable by COBRA. Clearly, it
is expected to be dependent on the observation conditions and generally speaking the best detection
limit is obtained for sites with low noise on the $SO_2$ SCDs and the highest measurement sensitivity
(i.e. high AMFs). These sites are found at low latitudes and in particular at high elevations or for
high albedos. To estimate the emission detection limit, we define the statistical significance of an
emission signal as three times its standard error. Based on the global sources presented above
(section 5.1), we performed statistics using this metric. To avoid biases by the strongest sources,
we only considered the sources with estimated emissions less than $50\,kt\,y^{-1}$. The resulting detection
limit values are found in the range between 4 and $11\,kt\,y^{-1}$ depending on the AMF, with a mean
value of $8\,kt\,y^{-1}$. It is important to realize that this limit of detection is remarkably low, at least
twice better than using TROPOMI DOAS data. It is also a factor of 4 smaller than the detection
limit of $30\text{-}40\,kt\,y^{-1}$ offered by OMI for the first years of operation (Fioletov et al., 2016; McLinden
et al., 2016). This suggests that the TROPOMI COBRA implementation is optimal in exploiting
the gain in spatial resolution of TROPOMI compared to OMI (~16 times smaller pixel sizes). This
finding is actually supported by the fact that the noise levels for individual pixels for TROPOMI
COBRA and OMI PCA VCDs are similar (not shown; see also section 3 of Fioletov et al., 2020).
In the following, we demonstrate the potential of the TROPOMI COBRA $SO_2$ data set to detect
and quantify weak emissions. For this, we use a slightly adapted version of the inversion technique
of section 5.1, and illustrate the method on a selection of new emission sources.
The region of interest is the Dhofar governorate in southern Oman. There, the exploitation of oil
and gas fields is growing fast, with a number of rapidly evolving projects of exploration and
production. In Figure 8 (left column), the yearly averaged TROPOMI $SO_2$ maps over South Oman
are shown for 2018-2020. One can clearly identify and isolate 3 main emission locations, namely
the Rabab Harwell integrated plant (18.03°N, 54.64°E), the Birba Gathering station (18.32°N,
55.10°E) and the Tayseer gas field (18.71°N, 55.34°E). We note that these emission sources are





not listed in any emission inventory and the actual locations of the sources are approximated from
available visible imagery (e.g. Google Earth). A noticeable feature in Figure 8 is the very low
observed $SO_2$ column level, in particular over Birba Gathering and Tayseer with $SO_2$ VCDs of 0.03
– 0.1 DU, reflecting again the great sensitivity of COBRA. To estimate the $SO_2$ emissions from
the TROPOMI data, the source method used in section 5.1 has been refined and tuned for this
particular case study. A multi-source $SO_2$ emission retrieval was applied as in Fioletov et al. (2017)
with one modification: a regression term proportional to the elevation was added to the fit to adjust
for a small altitude-related bias in retrieved $SO_2$ (the values were slightly lower over the mountains
near the Arabian Sea coast). This multi-source method is motivated by the fact that the sources are
close to each other (~ 50-100 km distance) and the emissions cannot be fitted separately. Here, the
approach basically allows for overlaps of the modelled $SO_2$ spatial distributions; the emissions
from the individual sources are then adjusted so that the total $SO_2$ modelled field fits best the
observed $SO_2$ VCD distribution. In Figure 8, the results of the fit are shown (center column), as
well as the residuals of the fit (right column). The estimated annual $SO_2$ emissions for the 3 sources
are given in the inset of Figure 8. Note that for this particular case, the emission detection limit (as
defined above) is typically of about 6 kt yr$^{-1}$. For the Rabab Harwell site, the algorithm retrieves
rather high and stable emissions over the years, with an average value of about 40 kt yr$^{-1}$, which is
well above the estimated detection limit. Interestingly, the Rabab Harwell site has large residuals
of ~0.1 DU for all years. This suggests that the point source representation used here is likely not
sufficient to explain the observations and it is possible that there are many small contributing
sources in the area. For the Birba Gathering site, the estimated emissions are much smaller and lie
in the range of 7-13 kt yr$^{-1}$. Yet, there is a good confidence that these emissions are real, given that
the estimates are a factor of 1-2 larger than the limit of detection. However, it is clear that the
uncertainty of the emission estimates are quite large. For the Tayseer site, an $SO_2$ signal could be
detected only recently. In 2019, the estimated emissions are of 2 kt yr$^{-1}$, i.e. below the detection
limit, and in 2020, the $SO_2$ emissions strongly increased to about 20 kt yr$^{-1}$, probably as a result of
a change in operation at the production facility. Finally, note that no significant residuals could be
found neither for Tayseer nor Birba Gathering site, and this suggests a point source behavior, at
both sites.

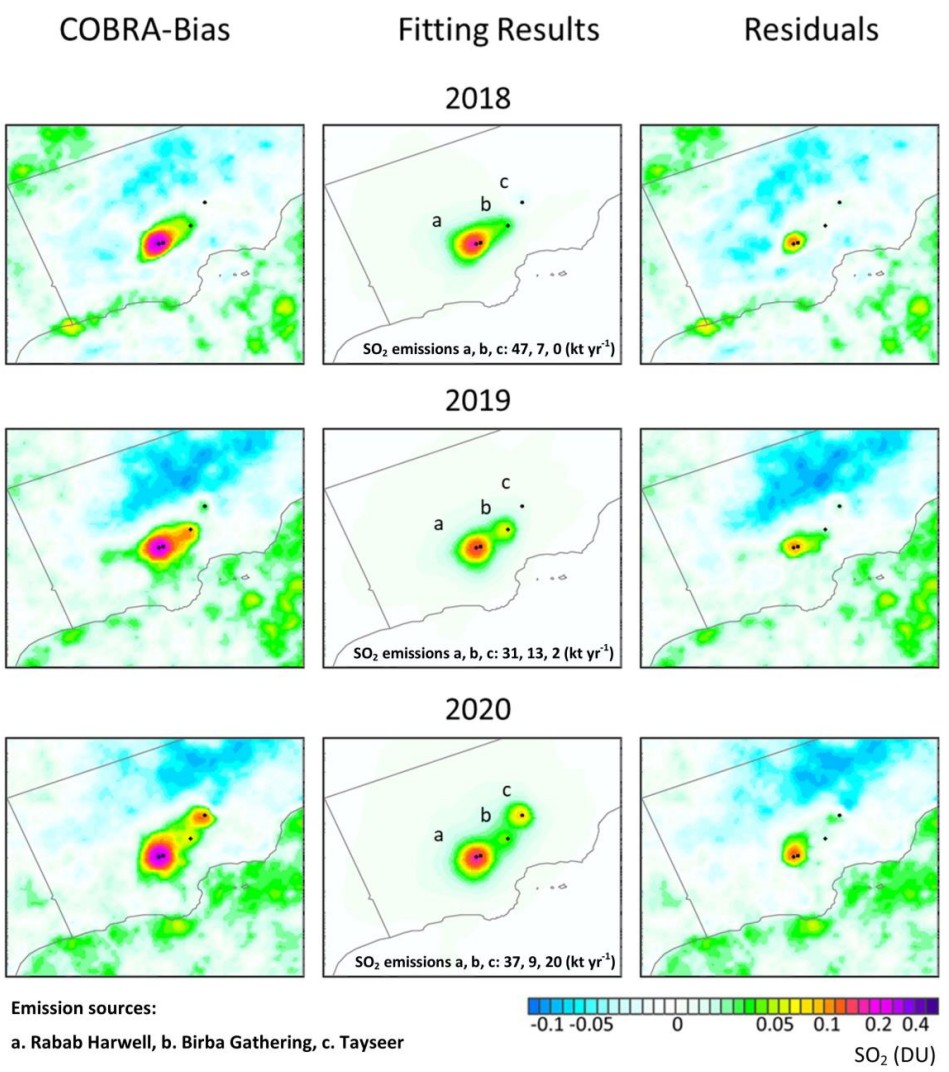



Figure 8. (Left) Yearly mean TROPOMI SO$_2$ columns retrieved from COBRA over South Oman for 2018 (April to December), 2019, 2020 (from top to bottom), after bias correction for the effect of elevation (see text). Three distinct SO$_2$ spots are discernable from the maps and are the results of emissions from oil and gas fields, referred as Rabab Harwell, Birba Gathering and Tayseer, (center) results of the fitting of TROPOMI SO$_2$ data. The estimated annual SO$_2$ emissions (given in the inset for the three sources) are used to reconstruct the SO$_2$ column field, (right) residuals of the fit or the difference between TROPOMI and fitting results.



In summary, the analysis over South Oman of Figure 8 nicely illustrates the strength of a highly sensitive scheme such as COBRA when applied to a high spatial resolution instrument as TROPOMI. The fact that such low $SO_2$ emissions can be tracked and quantified with that level of detail is remarkable. Although not shown, the emission inversion scheme was successfully applied not only to the South Oman sources but also to other test sites, as they were found in the global $SO_2$ map (Figure 6). The wind-rotation technique when applied to TROPOMI COBRA $SO_2$ data is arguably a promising tool to monitor weak $SO_2$ emissions and track the activity from rapidly emerging production facilities worldwide. However, applying the inversion scheme at the global scale is a significant effort, as it also requires some level of manual intervention and testing. For instance, the information on source type, location, etc. is typically lacking, and the supporting visible imagery - useful for identifying industrial facilities - is often outdated.

## 6. CONCLUSIONS

A new spectral fitting method for the retrieval of sulfur dioxide columns in the UV was presented and demonstrated for TROPOMI. Based on a dynamical total measurement error covariance, the method, called COBRA, allows reducing considerably the noise level (by a factor of 2) and biases present in the current TROPOMI DOAS $SO_2$ operational product. COBRA provides greater sensitivity to low $SO_2$ columns, and this conclusion is supported by MAX-DOAS observations. Preliminary comparison of COBRA to PCA retrievals suggests similar and even better algorithm performance. The $SO_2$ vertical column precision for individual pixel is in the range 0.5 - 1 DU; the systematic VCD uncertainty (contribution from the COBRA spectral fit only) is very small, typically less than 0.04 DU.

The benefit of COBRA is clearly demonstrated in this work using long-term oversampled averages. Owing to the excellent quality of the data (in terms of precision and accuracy), the high spatial resolution of TROPOMI can be better exploited. Zoomed $SO_2$ maps reveal new emission sources worldwide, with low $SO_2$ columns of 0.05 - 0.2 DU, or even lower.

By using the COBRA $SO_2$ data over large emission sources, we have recalculated the $SO_2$ emissions obtained by Fioletov et al. (2020) that were based on the TROPOMI operational $SO_2$ product. While the derived emission rates agree generally well, we found that the uncertainties on the emissions are significantly lower (up to 50%) using COBRA than with the operational product.





This opens the possibility to retrieve $SO_2$ emissions for weakly emitting sources, and we present a
number of examples that demonstrate the potential of the COBRA data in this direction.
With an estimated annual emission detection limit of about 8 kt yr$^{-1}$, the TROPOMI COBRA $SO_2$
data provides unique access to weak anthropogenic and volcanic point sources, and can help
completing current $SO_2$ emission inventories. It can also be used to track more accurately weak or
rapid changes in $SO_2$ levels, e.g., due to COVID-19 lockdown measures (Levelt et al., 2021) as
well as estimate seasonal and even monthly emissions. Finally, COBRA data would be particularly
relevant for the CAMS assimilation system as well.
COBRA is a good candidate for an implementation in the TROPOMI operational processor, with
limited computational resources and without the need for a separate background correction processor.
COBRA is also adaptable to other satellite instruments, including from geostationary platforms.
In particular, the European Sentinel-4 mission would likely benefit from a COBRA approach for
the retrieval of $SO_2$ columns, as the atmosphere will be sounded under unfavorable large
observation angles.
Future work could also be dedicated to the application of COBRA to historical sensors, in order to
produce a consistent long-term $SO_2$ data record, but also to the retrieval of other molecules.



**CODE AND DATA AVAILABILITY**
The TROPOMI COBRA $SO_2$ dataset is available from the corresponding author on request. The
TROPOMI DOAS $SO_2$ product is publically available on the Copernicus Sentinel-5P data hub
(https://s5phub.copernicus.eu). The TROPOMI PCA $SO_2$ dataset is available from Dr. Can Li on
request. The OMPS PCA $SO_2$ is publically available from Goddard Earth Sciences (GES) Data
Information Service Center (DISC)
(https://daac.gsfc.nasa.gov/datasets/OMPS_NPP_NMSO2_PCA_L2_2/summary).
The CAMS regional data are available from the Copernicus Atmosphere Data Store
(https://atmosphere.copernicus.eu/data/). The $SO_2$ emissions estimates can be obtained from Dr.
Vitali Fioletov on request. The MAX-DOAS measurements used to validate the satellite $SO_2$ data
are available on request from Drs. François Hendrick (Xianghe), Thomas Wagner, Vinod Kumar
(Mohali).
**AUTHOR CONTRIBUTIONS**
N.T. prepared the manuscript and figures with contributions from all the coauthors. N.T., I.D.S.,
C.Le., L.C., J.V., H.B., M.V.R. contributed to the development of the COBRA algorithm,
processing of the data and satellite comparison. V.F., C.McL., D.G. estimated the $SO_2$ emissions.
C.Li, N.K. developed the TROPOMI and OMPS PCA algorithms and provided data for the
comparison. P.H. and D.L. contributed to the development of the TROPOMI DOAS algorithm,
processing of the data and satellite comparison. A.I. and R.R. provided CAMS SO2 data. T. W.,
V.K., F.H, M.V.R. analyzed and provided MAX-DOAS data. All authors contributed to the
interpretation of the results and improvement of the manuscript.
**COMPETING INTERESTS,**
The authors declare that they have no conflict of interest.
**ACKNOWLEDGEMENTS AND FINANCIAL SUPPORT**
We acknowledge financial support from ESA S5P MPC (4000117151/16/I-LG), Belgium Prodex
TRACE-S5P (PEA 4000105598) and TROVA (PEA 4000130630) projects. We thank
EU/ESA/KNMI/DLR for providing the TROPOMI/S5P Level 1 products. This paper contains modified
Copernicus data (2018/2020) processed by BIRA-IASB.



C.Li and N.K. acknowledge support from the NASA Earth Science Division Aura Science Team, Suomi NPP Science Team and US participating investigator programs. L. C. is a research associate supported by the Belgian F.R.S-FNRS. We acknowledge Pucai Wang and Ting Wang (IAP/CAS, China) for their support in operating and maintaining MAX-DOAS observations in Xianghe. We acknowledge Dr. Vinayak Sinha for supporting us with the logistics to operate the MAX-DOAS at Mohali. V.K. acknowledges the Alexander von Humboldt foundation for supporting the postdoctoral fellowship.

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
