# Peer review of "A Sulfur Dioxide Covariance-Based Retrieval Algorithm (COBRA): application to TROPOMI reveals new emission sources"

_Atmospheric Chemistry and Physics, 2021_

## Author Comment (AC1)

**Review of Theys et al.: A Sulfur Dioxide Covariance-Based Retrieval Algorithm (COBRA): application to TROPOMI reveals new emission sources (https://doi.org/10.5194/acp-2021-294)**

We wish to thank the reviewers and editor for their positive and constructive comments, and time spent reviewing the manuscript. Each comment is addressed in detail below. Replies to comments are in blue; a list of references can be found at the end of the document; all figure numbers refer to the initial version of the manuscript.

**Comments of Anonymous Reviewer # 1**

In this manuscript, the authors report on the first application of COBRA, a new algorithm to the retrieval of $SO_2$ columns from measurements of the TROPOMI instrument. The algorithm is briefly explained and results for a full year of data are compared to columns from existing algorithms for $SO_2$ retrievals (DOAS, PCA). The performance of the new algorithm is further demonstrated by a comparison to modelled $SO_2$ fields (CAMS regional) and MAX-DOAS measurements in two locations. Long-term averages of the new $SO_2$ product are shown together with existing $SO_2$ source lists and emission estimates based on the new data are compared to those based on the operational product. Finally, an example is shown for a multi-source emission estimate of a weak source.

The topic of the manuscript fits into the scope of ACP although, in my opinion, it would have been a better match for AMT. The article is clearly structured and well written, the algorithm described and the nice results shown a clear improvement over existing data and certainly worth reporting, and overall, I have only minor comments and suggestions. The only general point I would like to make is, that this being the first report of the method, a more detailed discussion of the implementation and the tests performed would be appropriate.

Page 3, line 21: Not sure, if TROPOMI is the first mission with a tropospheric focus – I guess instruments like OMI, MOPITT or TES could also be seen as having this focus.

Agreed. The word 'first' is removed in the revised version of the manuscript.

Page 6, line 18: Maybe that is obvious, but can you please explain a bit more, what the difference is between the uncertainty in the $SO_2$ free spectrum and the measurement noise? Isn't in your method measurement noise one of the contributions to $\varepsilon_{bg}$?

The definition of $\varepsilon_{bg}$ as 'the uncertainty in the $SO_2$ free spectrum' is unclear. In Eq. 3, the term $\varepsilon_{bg}$ is simply the deviation of the SO2-free component of the spectrum relative to the mean spectrum $\bar{y}$. This is very different from the measurement noise $\varepsilon$ because $\varepsilon_{bg}$ includes atmospheric variability. Instead of retrieving $\varepsilon_{bg}$, the key of COBRA is to consider it as an error term. Doing so,

it is assumed that both errors can be grouped together as a total error ($\epsilon_{bg} + \epsilon$). The latter can be reasonably characterized in the form of a covariance matrix if a statistically representative set of SO2-free spectra is available. The text has been improved.

Page 7, line 5: Can you please elaborate on how equation 6 follows from equation 5?

The word 'follows' is not really adequate here, we have rephrased this part. Equation 6 corresponds to the square root of the error covariance of the solution, which reduces here to a single number given the fact that only one parameter is retrieved (i.e. $\widehat{SCD}$).

Page 7, line 25: How is wavelength calibration being dealt with in your method? Is there any analogue to shift and squeeze or are you assuming that wavelength calibration and stability of the spectra is so good that this is not needed?

This requires a more detailed discussion in the paper.

As pointed out in Section 2.2 (p5, l15), the algorithm makes use of wavelength calibrated spectra. To do so, the exact same approach is used as for the operational SO2 algorithm as described in Theys et al., 2017. In brief, a wavelength grid is determined by adjusting the reference solar spectrum Kurucz (degraded at TROPOMI resolution) to an irradiance spectrum as measured by TROPOMI. All spectral data (measured spectra and $SO_2$ cross-section) are then interpolated on this common grid. However, we feel that the detailed information on the wavelength calibration is not important in the context of the description of COBRA and we propose to simply mention Theys et al. (2017) for further details.

The reviewer is correct that for the analysis of the individual spectra, a common practice in DOAS is to fit a shift and squeeze. In principle, COBRA would allow to fit more parameters than SO2 alone but we argue it is preferable not do so (unless it is justified) because this would come with an increase of the SO2 data scatter/bias. Our justification also relates to previous work of Beirle et al (2013) that have shown that the effects of spectral shift and squeeze can be linearized and fitted as pseudo-absorbers. Therefore, their contributions (and variability) to the optical depth are essentially taken up by the covariance matrix. This is clarified in the text.

Please note that, in relation to the wavelength calibration raised by the reviewer, we have performed a test by switching off the wavelength calibration and we found that the TROPOMI COBRA results were mostly unchanged. This indicates that the wavelength assignment as reported in TROPOMI L1 is already very good.

Page 8, line 17: As this is the first report of an application of COBRA on UV/vis data, it would be good to add some discussion on the results of your tests and justification for the choice of parameters.

Indeed, this is the first paper of a UV/vis application of COBRA but on the other hand, it is not the first report on a measurement-based SO2 technique from space, as the OMI PCA SO2 algorithm was published by Li et al. (2013). We think that showing intermediate results (e.g. in SI) and entering into lengthy and technical discussion on the different tests (which consist mostly in 'trial and error' tests) would not serve the paper clarity and ACP scope. Therefore, we agree with the suggestion to justify better the choices of the parameters but in a succinct way. The text has been improved in this direction, wherever possible. Importantly, the main driver for the choice of the COBRA parameters was to facilitate the comparison with the PCA results. This has been clarified in the text.

Page 9, line 1: The need for $SO_2$ free spectra in each orbit, row, and latitude segment can be an important limitation of this method in the case of volcanic $SO_2$ plumes reaching the stratosphere. Please add some discussion on this point here, including some numbers on how many measurements you had to skip in your data set because of this constraint.

Yes currently this is a limitation of the method. Future algorithm versions, in particular for implementation in the SO2 processor, will include a better handling of this problem, e.g. by using a covariance matrix fallback constructed from previously processed orbits. However, for this first version, the amount of data skipped is modest, on the order of 0.025% in total. We have specified this in the text.

Page 10, line 13: Was the background correction applied by row? If so, why do we see the low-frequency variations in the results? If not, why not?

Indeed the background correction is applied by row. However, the mean background corrected SCDs are not equal to zero because the parameterization of the background correction is not constructed strictly based on SCDs coinciding with the equatorial Pacific sector used for Fig 1. The low frequency variation is not well understood and is also present for other trace gas products of TROPOMI (e.g. formaldehyde or glyoxal). Here we propose to clarify that the background correction is indeed applied by row and also that this low frequency (and unphysical) variation in the operational product is not well understood.

Figure 2: To make this a bit more quantitative, it would be good to add scatter plots between the different existing products and the new COBRA data.

We have generated the scatter plots proposed by the reviewer but the graphs are of little use. This is related to the fact that most of the grid cells are free of SO2, and the regression analysis are meaningless. One option would be to use a threshold on the SO2 columns but then the statistical parameters would depend on the actual cutoff value. Therefore we propose not to include the scatterplots.

Page 15, line 16: is likely not reflecting => is likely reflecting

We have changed the text.

Figure 5: It would be interesting to add similar figures for the operational DOAS product, maybe in the supplement

We agree with the suggestion. The figure is added in the supplement and copied below.

[Figure]

Compared to the COBRA comparison, the DOAS results are clearly worst both in terms of the correlation coefficients and slopes of the regression lines, for both sites (Xianghe and Mohali).

Page 19, line 1: I could not find any link or other means to access this file

During the submission, we have tried to add the file as supplement but it turned to be difficult. We will contact the Copernicus editorial office to see how to proceed.

Figure 7: What does the size of the markers in the left panel stand for?

The size is proportional to the ratio between the emission value to its standard deviation. The ratio was calculated for DOAS ($R_{DOAS}$) and COBRA ($R_{COBRA}$) and the size is proportional to their mean value, i.e. size= ($R_{COBRA}$+ $R_{DOAS}$)/2. This is now clarified in the figure caption.

Figure 7: On which of the two emission estimates is the size of the marker in the right panel based on?

On the average of the two. I.e., (DOAS_emissions+COBRA_emissions)/2. This is clarified in the figure caption.

Page 28, line 14: "fairly consistent" – this is a vague formulation! Why not check if the values agree within their reported uncertainties? Why not add error bars to the left panel of Figure 7? It is an interesting piece of information for users of the existing emission values whether they are still valid (within their uncertainties) or if numbers will change with the new product. If the latter is the case, this would warrant some discussion.

"fairly consistent" is just our description of the results plotted in Figure 7a.

On the scale of Figure 7a, the error bars will be about the size of the markers for sources with emissions >100 kt y$^{-1}$ increasing rapidly for sources with smaller emissions. They would make the plot very busy.

Instead of error bars in Figure 7a, we show the ratios of the emission estimate to its standard deviation. They are plotted in Figure 7b. The statistical error bars are small due to a very large number of individual pixels. They may be underestimated because possible correlation between errors of individual pixels was not considered.

The reviewer asked an important question about the validity of the previous estimates. To answer it, we calculated the differences between DOAS and COBRA emission estimates divided by the standard deviations of the DOAS-based emissions. The results are shown in the histogram below (the absolute numbers of sources in each bin are shown). For 87% of all sources, the differences are within ±5 standard deviations. And, for 66% of all sources, they are within ±3 standard deviations. These numbers are roughly twice bigger that would be expected if errors of individual pixels are uncorrelated (the standard deviation of the DOAS − COBRA emissions difference is sqrt ($\sigma_{DOAS}^2$+ $\sigma_{COBRA}^2$)). Large differences between DOAS and COBRA emission estimates for some sources are related to problems with the DOAS algorithm. For example, the 3 sources with the largest differences are in Iran where local DOAS biases are particularly large. We have clarified this in the text.

[Figure]

Page 28, line 18: For some of the emission estimates, COBRA has smaller ratios. Have checked why?

The uncertainties of emission estimates depend on standard deviations of errors for individual pixels and the number of these pixels. There is some difference in the number pixels available for the emission estimates between DOAS and COBRA as shown in Figure below. While the number of pixels is typically larger for COBRA, there are many exceptions. They may produce smaller overall emission uncertainties for DOAS even if the errors of individual pixels are larger for DOAS than for COBRA.

[Figure]

Page 29, line 17: I agree that this indicates that COBRA is good in exploiting the gain in spatial resolution provided by TROPOMI; if it is optimal in doing so I wouldn't know.

The reviewer is correct, the word 'optimal' is too strong and is removed in the revised mansucript.

**Comments of Neil Harris:**

This manuscript describes a new optimal estimation algorithm for UV SO2 which puts all the variability in the covariance matrix. It has been developed for use with TROPOMI data and shows reduced variability in the residuals as well as lower limits of detection. These improvements enable changes in source strength to be more readily observed and for weaker sources to be monitored. It is good work which should be published after minor corrections.

I agree with all Referee 1's comments and think they should be made.

It would be good if you can address the comment about ACP vs AMT. One way of making it more 'ACP' is to add a bit more on the interpretation of the new estimates of SO2 emissions and what the implications are for model studies using existing inventories. I was surprised to see that lower detection limits did not lead to more SO2 emissions being estimated overall. Does that mean even smaller sources are unimportant? Such a discussion would also strengthen the broader conclusions.

We only looked at the sources that were previously detected from OMI. This is why no additional sources are included and the overall emissions are not larger than previously estimated. The histogram below shows the distribution of the ratios between the standard deviation of COBRA-based emission estimate to that from DOAS. The most ratios are between 0.7 and 0.8, i.e., the uncertainties of COBRA-based emission estimates are about 30% lower. That means we potentially can see more sources using COBRA than DOAS. In sections 4 and 5.2, we demonstrate indeed the ability of COBRA to detect many new sources. However, we have neither identified and classified all new sources in the global COBRA data nor estimated the corresponding emissions, because this is a very significant amount of work, which goes beyond the scope of the paper. Yet, we think the paper illustrates well the potential of the data for future research on improved monitoring and quantification of anthropogenic and volcanic $SO_2$ emissions, and is therefore suitable for ACP.

[Figure]

Two other comments:

1.  How does the type of land surface, and particularly its spectral signature, affect the retrieval? You mention particular land surfaces in respect to a couple of examples of less good agreement. Is that related to a retrieval issue or to possible emissions from that land surface? Similarly for aerosol loading.

    Over the UV wavelength interval used for the retrievals, land surfaces have no specific spectral signatures that could interfere with SO2. Change in land spectral reflectance occurs essentially as an overall intensity change. As explained in the text, very dark/very bright scenes might be underrepresented by the spectra used to calculate covariance matrices and this can lead to offsets. Because of this, we think real emissions from land surfaces are unlikely to explain the observations.

    For aerosols, the same is true, as for the lack of specific spectral signature. For urban scenes, an aerosol layer can be seen as a cloud with a certain albedo, from the intensity point of view. By design, the covariance matrices cover a large range of conditions (in terms of cloudiness) and a bias, specifically related to the presence of aerosols, is not

expected. Note that for strong volcanic eruptions, scenes with large amounts of volcanic ash (UV absorbing aerosols) can be considered as intensity outliers and possibly lead to biases. However, these conditions are filtered out from the data.

2. Could the potential error sources / limiting factors be mentioned as well as the advantages? Is this the perfect algorithm which is limited by measurement characteristics?

   We agree with the referee, it would be better to present the pro and cons of the method. The conclusion section is modified to make this clearer, e.g. by emphasizing the conditions for which COBRA is performing less well (point 1 above). Although COBRA has many advantages, the presented algorithm is not the perfect algorithm either.

Minor comments

Page 29, line 19 – delete 'actually'

We have changed the text.

Page 30, line 11 – 'spatial distributions: the emissions'

We have changed the text.

**References**

- Beirle, S., Sihler, H., and Wagner, T.: Linearisation of the effects of spectral shift and stretch in DOAS analysis, Atmos. Meas. Tech., 6, 661–675, https://doi.org/10.5194/amt-6-661-2013, 2013.

- Li, C., Joiner, J., Krotkov, N. A. and Bhartia, P. K.: A fast and sensitive new satellite SO2 retrieval algorithm based on principal component analysis: Application to the ozone monitoring instrument, Geophys. Res. Lett., 40(23), 6314–6318, doi:10.1002/2013GL058134, 2013.
- Theys, N., De Smedt, I., Yu, H., Danckaert, T., van Gent, J., Hörmann, C., Wagner, T., Hedelt, P., Bauer, H., Romahn, F., Pedergnana, M., Loyola, D. and Van Roozendael, M.: Sulfur dioxide retrievals from TROPOMI onboard Sentinel-5 Precursor: algorithm theoretical basis, Atmos. Meas. Tech., 10(1), 119–153, doi:10.5194/amt-10-119-2017, 2017.